# Caveolae-Mediated Extracellular Vesicle (CMEV) Signaling of Polyvalent Polysaccharide Vaccination: A Host–Pathogen Interface Hypothesis [note 1]

**DOI:** 10.3390/pharmaceutics14122653

**Published:** 2022-11-30

**Authors:** Shengwen Calvin Li, Mustafa H. Kabeer

**Affiliations:** 1Neuro-Oncology and Stem Cell Research Laboratory, Center for Neuroscience Research, CHOC Children’s Research Institute, Children’s Hospital of Orange County, 1201 West La Veta Ave., Orange, CA 92868-3874, USA; 2Department of Neurology, University of California-Irvine School of Medicine, 200 S Manchester Ave. Ste 206, Orange, CA 92868, USA; 3Division of Pediatric General and Thoracic Surgery, CHOC Children’s Hospital, 1201 West La Veta Ave., Orange, CA 92868, USA; 4Department of Surgery, University of California-Irvine School of Medicine, 333 City Blvd. West, Suite 700, Orange, CA 92868, USA

**Keywords:** Polyvalent polysaccharide vaccine (PPSV), polysaccharides, caveolae, vaccine, signaling, caveolae memory extracellular vesicles (CMEVs)

## Abstract

We published a study showing that improvement in response to splenectomy associated defective, in regards to the antibody response to Pneumovax^®^ 23 (23-valent polysaccharides, PPSV23), can be achieved by splenocyte reinfusion. This study triggered a debate on whether and how primary and secondary immune responses occur based on humoral antibody responses to the initial vaccination and revaccination. The anti-SARS-CoV-2 vaccine sheds new light on the interpretation of our previous data. Here, we offer an opinion on the administration of the polyvalent polysaccharide vaccine (PPSV23), which appears to be highly relevant to the primary vaccine against SARS-CoV-2 and its booster dose. Thus, we do not insist this is a secondary immune response but an antibody response, nonetheless, as measured through IgG titers after revaccination. However, we contend that we are not sure if these lower but present IgG levels against pneumococcal antigens are clinically protective or are equally common in all groups because of the phenomenon of “hyporesponsiveness” seen after repeated polysaccharide vaccine challenge. We review the literature and propose a new mechanism—caveolae memory extracellular vesicles (CMEVs)—by which polysaccharides mediate prolonged and sustained immune response post-vaccination. We further delineate and explain the data sets to suggest that the dual targets on both Cav-1 and SARS-CoV-2 spike proteins may block the viral entrance and neutralize viral load, which minimizes the immune reaction against viral attacks and inflammatory responses. Thus, while presenting our immunological opinion, we answer queries and responses made by readers to our original statements published in our previous work and propose a hypothesis for all vaccination strategies, i.e., caveolae-mediated extracellular vesicle-mediated vaccine memory.

## 1. Introduction 

Viral envelope proteins and bacterial exotoxins trigger T cells to process antigens and present them to B cells, producing neutralizing antibodies against reinfection of *Streptococcus pneumoniae* (pneumococcus) [1]. Some of these B cells mutate the genes that encode their immunoglobulin variable regions through an ordered series of cellular and molecular changes in vivo and convert themselves to memory B cells, which respond to repeat antigen presentation. However, the mechanisms by which memory B cells recognize antigens are not fully elucidated. Based on the current literature related to our recent publication, we speculate on a new mechanism for adaptive immunity mediated through caveolae-mediated extracellular vesicles (CMEVs). 

Adaptive immunity occurs as memory B cells are activated to produce antibodies with hyper-responsiveness to repeated antigen exposure, related to invading micro-organisms in a T-cell-dependent manner. Some mechanisms have been described, but many are still unknown and require further investigation. We showed in [2], in Figure 1, that positive control mice without splenectomy exhibited a normal immune response. The splenectomized mice (experimental group) with splenic lymphocyte reinfusion had a higher IgG titer than those without reinfusion (negative control group). Our data could be explained through the phenomenon of “hyporesponsiveness” seen after repeated polysaccharide vaccine challenges [3]. We argue that administering a plain Polyvalent polysaccharide vaccine PPSV23 is highly relevant to the primary vaccine against SARS-CoV-2 [4] and the booster dose [5] due to potential subsequent glycosylation sharing similarities to the polysaccharide antigens. The mRNA-1273 vaccine can neither prevent infection nor the spread of SARS-CoV-2, but it can reduce the severity of COVID-19 with antibodies against SARS-CoV-2 variants at low levels for six months [6]. The booster shot of the vaccine can neutralize the SARS-CoV-2 Omicron variant (BA.1/B.1.1.529), consisting of 36 mutations in spike protein based on 239 human subjects [7]. All of these lead to a question: Why were there no secondary T-cell immune responses, but the booster vaccine still had efficacy?

Even though these discrepancies challenge the dogma that secondary immune responses must go through T cells, some suggest that we have only observed the antibody response without profoundly exploring the mechanisms, including the possible presence of specific memory B cells, the possible cooperation of other immune cells, as well as IgG subclass switching.

We chose Pneumovax^®^ 23 over Prevnar^®^ 13 because we initially only wanted to study the response based on isolated B cell activation from the vaccine. In addition, we thought that the immune titer from the first or second Prevnar vaccine would be much higher due to protein-conjugated T-dependent antigens and could potentially obscure the contributions from the B cell component. We believe this response needs to be investigated as a separate project. 

We have received questions and comments about the possible interpretation of our data: some readers argue that this is NOT a secondary immune response but an antibody response as measured through IgG titers after revaccination. Here, we wanted to revisit some historical data to explore this argument. 

One report shows that polysaccharides induce memory B cell differentiation to differ from immune responses related to protein antigens. A polysaccharide-specific immunoglobulin G (IgG) response results from memory B cells that act as T-independent (TI) type II immune responses in naive B cells sensitive only to polysaccharides [8]. There are two main types of lymphocytes: T cells and B cells. T lymphocytes, or T cells, are a significant component of the adaptive immune system. When a T cell encounters a recognizable APC, the naïve cell receives a signal to mature. There are three types of signals: TCR, BCR, and cytokine signals. If a cell gets all three signals, it becomes an effector cell. Effector cells are functionally divided, acting as (1) cytotoxic T Cells, also known as CD8+ cells, which have the primary job to kill toxic/target cells (infected host cells); (2) T helper cells, which produce cytokines and thereby stimulate B cells; (3) regulatory T Cells; and (4) memory T Cells. Polysaccharide antigens fail to be recognized and bound by the T cell antigen receptor complex; however, TI-2 antigens do not stimulate T cells, T cell helper, non-T cells (e.g., NK cells), or cytokine production as IL-3, GMCSF, and IFN-gamma are known to interact with B cells [9]. 

Polysaccharide antigens fail to be recognized and bound by the T cell antigen receptor complex; the immune response receives no contributions from T cells. A knowledge gap, thus, exists that polysaccharides create memory B cells that are different from the mode of action of protein antigens. Polysaccharide-specific immunoglobulin G (IgG) governs memory B cells that act as T-independent type II immune responses in naive B cells sensitive only to polysaccharides. We postulated that polyvalent polysaccharide vaccines acting via caveolae-mediated memory extracellular vesicles (CMEVs) result in prolonged and persistent signaling to B cells. The CMEV hypothesis refers to the absence of added cell-mediated stimulation. The hypothesis that CMEV-mediated cytokines drive up the in vivo responses to TI-2 antigens is supported by retrospective data in the literature.

Another report shows that memory B cells involved in the secondary response proliferate and differentiate like B cells in the primary response reaction without somatic hypermutation through the clonal selection theory [10]. B cell differentiation (Decision-making) switches based on either cell autonomy or the local environment. Nevertheless, how a B cell is driven into a particular differentiation pathway is highly relevant to our discussion.

Specifically, we could neither dispute nor prove how the microenvironment derived from splenectomy affects how memory B cells may develop upon antigen presentation by T helper cells in our prior publication since this was not part of the initial research study. We contend that we are uncertain if these lower but present IgG levels against pneumococcal antigens are clinically protective or are equally common in all groups because of the phenomenon of “hyporesponsiveness” seen after repeated polysaccharide vaccine challenge. 

The phenomenon of “hyporesponsiveness” came into the debate of controversies in the history of literature. In fact, as one reader has criticized, the long-standing knowledge that administration of a plain 23-valent unconjugated pneumococcal Polyvalent polysaccharide vaccine is not, by definition, linked to the possibility of a secondary immune response. Indeed, many other references support many aspects of this study. There is no doubt that protein-conjugated vaccines cause robust responses and yield more effective outcomes because they involve T-dependent responses to activate B cells through a different mechanism. However, the assertion that there is no stimulation of memory B cell activity through the revaccination using strictly polysaccharide vaccines (antigens) is contentious and perhaps false. An article published by the University of Oxford looked at this question and concluded that the responses to the protein-conjugated polysaccharide vaccine are significant, but that strict polysaccharide vaccines with no T-dependent activation can still induce memory B cell responses [11]. Their study noted, after a six-month post-primary vaccine period titers with PCV7, a memory B-cell helper response to the PCV7 booster vaccine. In contrast, booster PPSV23 responses were significantly lower than their counterparts with PCV7 for all serotypes except serotype 14, as they pointed out. Similar results were obtained with the 10-valent or the 13-valent vaccines [12]. These prompted us to consider several other points. 

First, other laboratories have identified specific splenic B cell populations using directed idiotypic expression of receptors against a pneumococcal antigen, and gathered metrics on the degrees of isotype mutation and somatic hypermutation and memory B cell production specifically attributed towards the unique reservoir of splenic lymphocytes. There seems to be protection for humans against specific infectious agents [13]. As shown in Figure 1 (Group B mice), the increased antibody production might not be driven by the booster dose but by a ride-on from the primary vaccination that peaks at week 7 [14]. An argument is that, in some mouse models vaccinated with pure T-independent antigens, such as PPSV23-polysaccharides, no secondary response is observed due to the lack of memory T cells. 

Emerging evidence shows a secondary immune response upon contact with an antigen. The question remains as to how it occurs. What is the mechanism of action? Previous studies showed that polysaccharides activate memory B cells differently from protein antigens [8], implying a novel mechanism. 

Secondly and alternatively, the partial correction of the splenectomy-associated defective antibody response to Pneumovax23 by splenocyte reinfusion may be due to memory B cells producing a secondary reaction, as triggered by the splenectomy, and its impact on decision-making involved in cell autonomy versus local environment [10]. We hypothesize that such a local microenvironment may cultivate caveolae-derived memory extracellular vesicles (CMEV) to carry the message of memory B cells to others within the body. This notion might explain why there are no secondary T-cell immune responses to PPSV23-polysaccharides. However, it still appears efficacious without needing T cell responses.

Thirdly, we believe that our observation of IgG response to the vaccine could be continued as a secondary immune response after the initial antigen exposure. This point is not crucial for the study’s final message, which is the partial correction of the splenectomy-associated defective antibody response to Pneumovax23^®^ by splenocyte reinfusion. We have only observed the antibody response (within the limits mentioned above) without exploring the mechanisms more deeply by looking for the possible presence of specific memory B cells, the possible cooperation of other cells, or the class and subclass switching.

We chose Pneumovax^®^ 23 over Prevnar^®^ 13 because we initially only wanted to study the response based on isolated B cell activation from the vaccine. In addition, we thought that the immune boost from the first or second Prevnar vaccine would be much higher, resulting from protein-conjugated T-dependent antigens and potentially hiding what might have happened just from the B cell component. We believe this response would be a necessary part of the study but chose to do this as a later project, rather than the initial one. Our initial goal was to investigate the specificity of the B cell component in the spleen and its ability to raise an immune response to Pneumovax^®^ 23. Our project was rooted in antibody recognition [15], host–bacteria system [16], and epidemiology [17] that eventually led to the currently utilized protein-conjugated Haemophilus influenzae vaccine. Below, we offer an opinion on the administration of the plain Polyvalent polysaccharide vaccine against PPSV23, which appears to be highly relevant to the primary vaccine against SARS-CoV-2 and its booster dose.

## 2. Speculation from New Perspectives on SARS-CoV-2 Vaccines

At a pivotal moment in history, scientists have learned from 1.7 billion doses of SARS-CoV-2 vaccines [18] what impact the vaccines have had on the course of the pandemic based on 40,000 people, with 95% effectiveness in protecting recipients from symptomatic COVID-19. How long does the protection against the disease last? A study of more than 25,000 healthcare workers in the United Kingdom found that a SARS-CoV-2 infection reduced the risk of catching the virus by 84% for at least seven months [19]. Memory B cell repertoire was identified in those who took the triple vaccinees against diverse SARS-CoV-2 variants [20]. More questions remain to be elucidated: (1) How much do the vaccines block transmission? (2) What is the underlying biology of long COVID? (3) How does long COVID affect post-infection syndromes? Even though the adjuvanted RBD–NP vaccines (NCT04742738 and NCT04750343) manifest against SARS-CoV-2 [21], these results open up a new perspective on vaccination relevant to our study.

### 2.1. The Hypothesis 

Cells secrete extracellular vesicles (EVs) into the extracellular space for communications. Based on the biogenesis, release pathways, size, content, and function, EVs are defined into microvesicles (MVs), exosomes, and apoptotic bodies [22]. Extracellular vesicles contain lipids, proteins, metabolites, DNA, RNA, and ncRNA that can mediate communication within host–pathogen interactions, and are evolutionarily conserved [23]. For example, the gastrointestinal nematode *Heligmosomoides polygyrus* has three EV production modes associated with parasitic infection [23], controlled by extra-cellular and intracellular parasites and dendritic cells. “The EVs by extra-cellular parasites are internalized by macrophages, which decreases IL-6, TNF, Ym1, and RELMα and downregulates IL-33 receptor subunit ST”. Further studies show that in vivo mice vaccinated with EV-alum can generate protective immunity against parasite challenges [24], suggesting that EV plays a role in immunity. *Mycobacterium tuberculosis* (Mtb) associated with tuberculosis (TB) might release EVs containing lipoglycans and lipoproteins [25]. EVs are closely related to the plasma membrane microdomain lipid rafts [26] and caveolae, indicating dynamic changes underlying physiological and pathological conditions. 

Viruses enter the plasma membrane microdomain (lipid rafts and caveolae) that contains virus recognition receptors and start the replication cycle, evolving in the endoplasmic reticulum (ER) [27], in the Golgi complex [28], on the membrane of endosomes [29], and in caveolae-associated flotillin-containing phagosomes [30]—all which represent the continuity of the same original domain. Within plasma membrane microdomains lie both “planar lipid rafts” and “caveolae,” consisting of flotillin and caveolin (Cav-1, -2, -3), respectively [31]. Note that the membrane is inherently dynamic, and membrane microdomains can adjust and adapt the gap between the “planar lipid rafts” and “caveolae” as needed. The other essential structural components of caveolae proteins include Cavin1 (PTRF), Cavin2 (SDPR), Cavin3 (PRKCDBP), the muscle-specific isoform Cavin4 (MURC), EHD2 [32], and pacsin (also termed syndapin) [33]. Responding to cellular stress and during endocytic trafficking trigger the reorganization of membrane microdomains mediated by the interaction of cavins and caveolin molecules within the caveola formation [34]. 

Recent advances in engineered nanoparticles have fueled the enthusiasm for the potential to improve cancer treatment and vaccination delivery using internalization pathways [35]. However, the discrepancy in clinical trials demands further studies. We found that “Single-cell transcriptomics uncovers the impacts of titanium dioxide nanoparticles on human bone marrow stromal cells,” which was not found in bulk cell populations [36]. Insights have been offered in an expert review [37]. “We have highlighted the variation in endocytic pathways in different commonly used laboratory cell lines. It is clear that there will be even more variation in endocytosis between different cell types in vivo, reflecting the particular properties of these cells, their physiological functions, and the ever-changing local environment of cells in different tissues within a whole organism. The organization of the endosomal circuits differs between cell types in vivo, and particular cell types, such as cells of the kidney proximal tubules, have evolved high-capacity internalization mechanisms. Even cells of the same type grown in culture under different conditions can dramatically remodel their endocytic pathways as they change from a dividing to a quiescent state. So it is not surprising that the few studies that have compared endocytosis by cells in culture with their in vivo counterparts have shown notable differences in endocytic pathways”.

Most PPSV vaccine efficacy is sustained through booster shots, yet significant knowledge gaps exist. To fill the knowledge gaps, we hypothesize that caveolin-1 (Cav-1)-mediated memory extracellular vesicles (CMEVs) carry the message around the bloodstream and provide an immune boost different from the primary or secondary antigen exposure.

EVs have been hypothesized to be regulated by Cav-1, a well-known structural protein of caveolae [38]. Cav-1 plays a critical role in Cav-1-mediated signal transduction through the Caveolin-Scaffolding-Domain (CSD) [39] that interacts with a CSD-interacting motif (CIM) among the partner receptors [40] (Figure 2). However, specific infection-initiated mechanisms remain elusive (Figure 3).

### 2.2. Cav-1/Caveolae-Budded EVs as Potential Hijackable Gates in Cell Communication of Caveolin-Related Diseases Named Caveolinopathies

Caveolae are present in membrane microdomains involved in endocytosis, transcytosis, cell signaling, mechanotransduction, and aging. Caveolae interface with the extracellular environment (Figure 2). Thus, some raise the hypothesis that caveolae could serve as hijackable “gatekeepers” or “gateways” [43]. Only caveolin-1 is considered a possible disease biomarker in clinical trials utilizing its scaffold proteins. 

### 2.3. Supporting Background: A Lesson from the COVID-19 Pandemic

Basic research has focused on specific infectious diseases, which derailed efforts to achieve a new uniform theory for all vaccination strategies until the world faced the COVID-19 pandemic. 

Indeed, the US Food and Drug Administration (FDA) approved clofazimine—an anti-leprosy drug to block the fusion event of viral spike glycoprotein and host ACE and viral helicase [44]. *Staphylococcus aureus* released EVs have been adopted as a vaccine platform [45]. Another example, caveolae-based EVs, can mediate the interaction of SARS-CoV-2 spike protein’s RBD with ACE2 [46] localized in caveolae (Figure 4); thus, SARS-CoV-2 spike-based vaccine mRNA-liposomes likely fuse with caveolae microdomain to enter the cell. 

#### Lessons Learned from COVID-19-Related Vaccination

COVID-19, caused by SARS-CoV-2 infection, raises questions about the process of viral transmission, which is related to how antiviral therapeutics are designed to enhance memory against viruses [47]. As shown in Figure 5, in lymphoid tissues, dynamic changes manifest in both virus-specific cytotoxic CD8+ T cells and CD4+ T helper cells, which transduce B cells to produce and release virus-specific immunoglobulin M (IgM) antibodies and prompt B cells’ class switching from IgM to IgG or IgA virus-specific and subset virus-specific memory, i.e., IgG/IgA B cells and T cells remain. CD8+ T cells specify effector cells to remove infected cells to cut off the viral replication factory. CD4+ T cells specify T helper 1 (TH1) cells to suppress viral replication directly. CD4+ T cells specify T follicular helper (TFH) cells to prompt antibody-producing B cells (plasma cells) and memory B cells. These antibodies block viral entry.

T- and B-cells are a critical factor in fighting off viral infections (Figure 5) via repertoires of T cell receptor (TCR) somatic hypermutation and clonal expansion and by B cell receptor (BCR) antibody class switching, such as IgM, IgG, and IgA subtypes. We have yet to understand children’s and adults’ adaptive immunity mechanisms fully. A possible link might be the EV–virus interface in establishing or boosting T and B cell antiviral immunity [48]. 

Both virus-specific cytotoxic CD8+ T cells and CD4+ T helper cells transduce B cells to produce and release virus-specific immunoglobulin M (IgM) antibodies and prompt B cells’ class switching from IgM to IgG or IgA virus-specific and subset virus-specific memory. IgG/IgA B cells and T cells remain (adapted from [47]). [Figure reproduced from [47] with the permission of Science-the AAAS Publishing].

SARS-CoV-2 patient-specific IgG and IgM are derived from S1- and N-proteins, ORF9b and NSP5 [49]. In innate and adaptive immunity, severe COVID-19 differs from non-SARS-CoV-2-driven pneumonia, particularly in T cell exhaustion exclusive to COVID-19, while circulating NKT cells measure patient outcome [50]. SARS-CoV-2- signature peptides bound to the HLA are incorporated into EVs for immunopathologic delivery at the viral action loci. 

The third dose of mRNA booster vaccines in a longitudinal cohort of 42 volunteers (either the Moderna (mRNA-1273; n  =  8) or Pfizer-BioNTech (BNT162b2; n  =  34) mRNA vaccine) (1) increased the frequency of Omicron RBD-binding which induced a diverse memory B cell repertoire, and (2) boosted plasma neutralizing antibody (nAb) responses to multiple SARS-CoV-2 variants, including Omicron [51]. Both memory B cells and nAbs were insufficient to prevent breakthrough infection in many individuals. However, it could help explain why the third dose of SARS-CoV-2 mRNA boost is highly protected from more severe consequences of infection, even though the mechanism is not fully characterized. A recent article reported that, in 7491 critically ill individuals, there were associations with the dysregulation of interferon signaling (IL10RB and PLSCR1), leucocyte differentiation (BCL11A), blood-type antigen secretor status (FUT2), myeloid cell adhesion molecules (SELE, ICAM5, and CD209), and the coagulation factor F8, as well as reduced expression of a membrane flippase (ATP11A) and increased expression of mucin (MUC1), compared to the 48,400 controls [52]. The governing mechanism is not fully characterized.

## 3. Evidence for Caveolae-Mediated PPS Signaling in Vaccination and Immunotherapy: Cav-1 Exalts Dual Roles of Pro-Inflammation and Anti-Inflammation upon Bacterial Internalization 

### 3.1. Caveolae Act as an Entry-Port of Infectious Agents and as a Signaling Organelle 

Caveolae, plasma membrane invaginations of 60–80 nm in diameter, are a subset of lipid rafts enriched in cholesterol and sphingolipids. Caveolin family proteins, including caveolin-1 (Cav-1), caveolin-2 (Cav-2), and caveolin-3 (Cav-3), are expressed in various tissues and cell types, such as endothelial cells [53]; smooth muscle cells [54]; cerebral artery-associated smooth muscle (SMCs) and endothelial cells (ECs) [55]; skeletal muscle [56]; cardiomyocytes [57]; glioblastoma [58] alveolar macrophages [35] neutrophils [59] and adipocytes [60]. Caveolae function in endocytosis, transcytosis, pinocytosis, and calcium signaling [26] and regulate various signaling events. “Addressing inconsistencies in caveolae-sepsis-regulated signaling pathways, including LPS, eNOS, and TLR4. NFκB, MKK3/p38 MAPK, cPLA2/p38 MAPK, STAT3/NFκB, IL-1β, and IL-1R1” [61] is complex and warrants further studies.

Wang and colleagues found that LPS binds to CD14 (a receptor for LPS) that is associated with the caveolae of monocyte-macrophage (THP-1 cell) [62]. These domains are discrete regions of the monocyte-macrophage plasma membrane, containing Cav-1, p53/56lyn, GTP-binding proteins, ouabain-inhibitable Na+/K+ ATPase [63], sphingomyelin, and GM1 ganglioside, to uptake molecules that bind CD14 [64]. 

Cav-1 expresses differentially in lipopolysaccharide (LPS) normal responsive (LPs(n)) fibroblasts compared to hyporesponsive (LPs(d)) fibroblasts, as in primary thioglycolate (TG)-elicited C3HeB/FeJ peritoneal macrophages, showing that the upregulation of Cav-1 is more substantial in LPs(d) than in LPs(n) fibroblasts upon stimulation with 1.0 pg of LPS/mL [65]. Caveolae take up invasin SfbI of group A streptococci [66]. Alternariol (AOH), mycotoxins produced by *Alternaria alternata* fungi, activate THP-1 macrophages and lipopolysaccharide (LPS) downstream through a pro-inflammatory signaling cascade [67]. Caveolae-mediated bacteria-uptake triggers inflammation through LAPF [68]. The interaction of pneumococcal surface protein PspC with the polymeric Ig-receptor (pIgR) in the host cells [69] triggers endocytosis via caveolae through Caveolin-1 Scaffolding Domain (CSD) [39] via CSD-interacting motif-containing molecules [40] and clathrin-coated vesicles [70]. Upon endocytosis, the engulfed pneumococci are routed into early endosomes for enzymatic degradations, resulting in the products pneumococcal polysaccharides (PnPS), which activate the host immune response [71]. Caveolin-1 (Cav-1) regulates the proinflammatory cytokine production induced by lipopolysaccharide (LPS) involving polymorphonuclear leukocytes (PMNs) [72]. 

### 3.2. Caveolae Bud out Extracellular Vesicles (EVs)

Given the above literature, we hypothesize that the reperfusion of salvaged autologous splenic lymphocytes might mediate extracellular vesicles (EVs) for immunity in the splenectomized host, rather than directly at the cellular level after splenic lymphocyte autotransplantation. We wanted to determine if reinfusion of autologous splenic lymphocytes could improve antibody titers to Pneumovax-23^®^ in a splenectomized mouse model. The results indicate that the reinfusion of autologous splenic lymphocytes might help patients’ immunity against post-splenectomy infection.

It is of interest to notice in the human data that both 10-valent and 13-valent pneumococcal conjugate vaccines (PCV10 and PCV13) elevate IgG levels (van Westen et al., 2018), [12]. In the Netherlands, either PCV10 or PCV13 was used to vaccinate 2-, 3-, and 4-month-old infants; they were measured for serum IgG antibody levels at 5, 8, and 11 months (booster dose). The authors found that PCV10 and PCV13 trigger antibody production at constant high levels after the primary vaccination for up to 11 months. Most importantly, PCV10’s 4/19F and PCV13’s 4/6B sufficiently trigger IgG seroprotection levels of IgG 0.35 μg/mL at 5-month and 8-month post-vaccination levels, respectively. Both PCV10 and PCV13 are polysaccharide-conjugate vaccines. As shown in Figure 1 (Group B mice), the increased antibody production might not be driven by the booster dose but by a ride-on from the primary vaccination that peaks at week 7 (Figure 1). An argument is that, in some mouse models vaccinated with pure T-independent antigens such as PPSV23-polysaccharides, no secondary response is observed due to the lack of memory T cells. However, such mechanism discrepancy remains to be fully elucidated in these splenectomized mice at the molecular, cellular, and organ levels. 

It is intuitive that the protection comes from a secondary immune response upon contact with antigens derived from a pneumococcal bacterial infection in patients with plain polysaccharide vaccines, such as PPSV23, to initiate signal transduction. How that happens remains elusive, prompting us to propose a novel mechanism (Figure 6). We speculate that the signal in question (with a question mark in Figure 6) is carried by exosome or microvesicles, including caveolae-mediated extracellular vesicles (EVs), as proposed here.

### 3.3. Cav-1 Mediated the Anti-Inflammatory Impact

The Cav-1/CD26 axis mediates the down-regulation of inflammation in macrophages [74]. DPP-4 inhibitors (teneligliptin) modulate Cav-1 association with CD26, down-regulating the TLR4/IRAK-4/TRIF/ERK pathway in macrophages. Toll-like receptor (TLR)4 also involves Cav-1 mediated mmLDL-CD14-induced spreading of human macrophages by oxPAPC-inhibited LPS signaling of inflammation in septic shock [75]. The Gram-negative bacterium produces lipopolysaccharide (LPS) complexes with TLR4, MD-2, and CD14 to initiate an inflammatory response in innate immune systems [76]. This inflammatory response is abolished by filipin, a sterol-binding agent. We have shown that sterol is essential in caveolae formation [77], and filipin’s disruption of sterol structure collapses caveolae [2], which indicates that the caveolae-dependent endocytosis pathway down-regulates LPS-induced TLR4 and MD-2 through membrane-anchored CD14. 

LPS activates Cav-1-expressed B-lymphocytes, which are essential in developing thymus-independent immune responses [78]. Additionally, in response to lipopolysaccharide (LPS), macrophages lacking cav-1 increase the production of toxic mediators that render Cav-1(−/−) mice sensitive to *S. enterica* serovar infection [79].

In Cav-1(−/−) mice, LPS elevates neutrophil sequestration (16×), lung microvascular permeability K(f,c) (5.7×), and edema formation (1.6×) in sepsis, resulting in endothelial NO synthase (eNOS)-derived NO production [80], which confirms that Cav-1 negatively modulate eNOS [81], similar to LynTK, SrcTK, G-proteins, and Ras signaling [39].

A downstream effector Janus kinase (JAK)-2 is regulated by endotoxin [lipopolysaccharide (LPS)] in life-threatening proinflammatory response (PR) that triggers severe growth hormone (GH) resistance, suggesting a unique discrete signal transduction switching mechanism through which Cav-1 interfaces with (1177)phospho-SER-endothelial nitric oxide synthase for nitration at the (1007)Y-(1008)Y of JAK2 [82].

Designers can take a synthetic analog of the dsRNA virus packed with *E. coli* lipopolysaccharide (LPS) in liposomes, which are taken through caveolae-dependent endocytosis onto macrophage plasma membranes, provoking innate immunity pathways in both zebrafish hepatocytes and trout macrophages [83]. 

### 3.4. Cav-1 Mediated the Pro-Inflammatory Impact

LPS-mediated CD14 activates Src-Tyrosine kinase to phosphorylate caveolin-1 at Tyr in endothelial cells (14), Cav-1 specific [84].In sepsis-induced lung inflammation and injury, LPS triggers the association of Cav-1-pTyr14 with TLR4, thereby inducing NF-kappa B signaling for proinflammatory cytokines [85].

In pulmonary arterial hypertension (PAH), when mutations in the last 21 amino acids of Cav-1 are replaced with a novel 22-amino-acid sequence (Rathinasabapathy et al., 2020), an inflammatory challenge [low-dose lipopolysaccharide (LPS)] leads to metabolic deficiencies and mild pulmonary hypertension in mouse models.

Macrophages mediate P2X7 receptor-based pro-inflammatory signaling, as modulated by ectonucleotides (CD39) in caveolae and lipid rafts [86]. This LPS-initiated ATP priming down-regulates STAT3 caveolin-1-deficient macrophages. 

SOD-bound antibodies against caveolar Plvap lead to CD31 Ab/SOD-related LPS signaling [87].for pro-inflammatory signaling mediated by reactive oxygen species. 

Lipopolysaccharides trigger E3 ubiquitin ligase ZNRF1 to act on caveolin-1 ubiquitination and degradation, activating TLR4. TLR4 activation modulates Akt-GSK3β through the ZNRF1-CAV1 axis to enhance pro-inflammatory cytokines and inhibit anti-inflammatory cytokine IL-10. Mice with deletion of ZNRF1 in their hematopoietic cells display increased resistance to endotoxic and polymicrobial septic shock due to attenuated inflammation. Our study defines ZNRF1 as a regulator of TLR4-induced inflammatory responses and reveals another mechanism for regulating TLR4 signaling through Cav-1 [88]. We do not know if caveolae-mediated PPS signaling in vaccination and immunotherapy is cell-type specific. Neither do we know how anti-inflammatory or inflammatory action switches in those involved cells. 

### 3.5. Beyond Caveolae: Engineered-EVs

Researchers have engineered mammalian extracellular vesicles (EVs) to deliver a host IFITM3 to decrease ZIKV infection [89]. EVs shuttle viral and host factors to modulate viral infection and transmission [90]. Thus, manipulating EVs can induce an anti-cancer immune response and deliver therapeutics [91]. EVs secreted by glioblastoma can activate dendritic cells for DC-SIGN-mediated CD4+ and CD8+ T cell activation for tumor suppression [92]. 

These lines of evidence show that exosomes are associated with a large spectrum of diseases, including cancer, cardiovascular disease, CNS, and viral infection [42]. However, little is known about exosome-mediated antiviral immunity. A new report shows that exosomes modulate ISKNV in mandarin fish [93], as analyzed using mass spectrometry in in vitro and in vivo studies. Another report shows that MSC-mediated EVs regulate the pathogenicity of influenza viruses [94].

In contrast, immune cells crosstalk innately to adaptive immunity through EVs in the cancer-immunity cycle to modulate cancer progression and metastasis [95]. Nevertheless, in feedback, tumor cells admit the tumor antigens, HSP70 and HSP90, via EVs to regulate Fas-L, TGF-β, and PGE2, which give rise to novel therapeutic EVs [96]. These issues demand both the ISCT and the ISEV provide guidelines on EV applications [97].

## 4. Therapeutic Intervention Involving Caveolae-Mediated Vehicle of Therapeutic Delivery Approaches to Target the CXCL12/CXCR4/CXCR7 Axes

Cell membrane-associated receptors, upon agonist–receptor interactions, activate a signaling cascade in physical contact with downstream signaling molecules, depending on the membrane’s lipid composition, which renders the changes in the regulation of lipid bilayer-derived EVs a novel therapeutic strategy [98]. We called these changes “EV switches” as they alter the cell’s status (e.g., proliferation, differentiation, death) in response to the modulation of EVs (Figure 6). Indeed, this discovery enables membrane-lipid therapy or multitherapy (https://doi.org/10.1016/B978-0-323-42978-8.00008-5, 27 August 2022).

Caveolae are clustered with caveolin proteins (Cav-1,2, 3), high cholesterol and sphingolipid [77], Src-tyrosine kinases [39] heterotrimeric G-proteins [99], and endothelial nitric oxide synthase (eNOS) [81]. Dextrin-mediated cholesterol depletion activates eNOS phosphorylation at Serine (1177) and disrupts caveolae through an upregulation of reactive oxygen species (ROS) in human umbilical vein endothelial cells (HUVECs), which are associated with hypertension [100]. However, inflammatory and metabolic defect-associated pulmonary arterial hypertension were observed in the expression of a human caveolin-1 mutation in mice [101].

Ferritin-based nanocarriers of superoxide dismutase (SOD) with uniform size (20 nm diameter) can target caveolar plasmalemmal vesicle-associated protein (Plvap) in endothelial cells to modulate pro-inflammatory cytokines and lipopolysaccharide (LPS) or anti-inflammatory effects [102]. All of these features make caveolae an ideal drug carrier (https://doi.org/10.1016/B978-0-323-42978-8.00008-5, 27 August 2022).

Along this line, the designer vaccine liposomal LPS-dsRNA activates innate immunity for pro-inflammatory and anti-viral action in zebrafish and trout macrophages [83]. Transcription-regulated miRNA-15a carried by cationic cyclodextrin-graphene nanoplatform (GCD@Ada-Rhod) activates the caveolae-mediated endocytosis internalization pathway without significant cytotoxicity (DOI: 10.1021/acsami.9b15826), and may conjure up hope for cancer therapy that can regulate the oncogene protein BCL-2 level [103]. *Rehmannia glutinosa* polysaccharides derived from PEGylation nano-adjuvants stimulate macrophages to secret pro-inflammatory cytokines through macropinocytosis-dependent and caveolae-mediated endocytosis to boost immunity [104]. Liposomal adjuvants promote biodistribution in the lymph nodes, leading to the innate immune system. Specific immunostimulatory agents can further potentiate the immunogenic properties of nucleic acid-based vaccines [105]. Computationally designed proteins drive the clustering of antibodies in multivalent assemblies against many targets, such as antibody nanocages against the SARS-CoV-2 spike protein, which contains the antibody and its binding homo-oligomer [106] with enhanced antibody-dependent signaling and efficacy of neutralization of the viruses. 

Similar approaches to the pneumococcal polysaccharide vaccine (PPV) enhance the primary prevention of acute coronary syndrome (ACS) in the elderly in Australia, thereby reducing cardiovascular disease [107]. Recently, we found that titanium dioxide nanoparticles (TiO_2_ NPs) might act on caveolae in human bone marrow stromal cells to change the molecular profiles based on sensitive single-cell RNA-seq (scRNA-seq) transcriptomes [108] with CRIF1 as a potential target [36]. This finding might help reduce TiO2 NP-associated toxicity. Both BNT162b2 and mRNA-1273 vaccines have substantially decreased morbidity and mortality, leading to different humoral immune responses to SARS-CoV-2. The mRNA-1273 recipients were found to produce stronger epitope-specific antibodies against the receptor binding domain (RBD) and N-terminal domain (NTD) than the BNT162b2 recipients [109]. Omicron (B.1.1.529), the most frequently mutated SARS-CoV-2 variant, evolves more than 30 mutations in its spike (S) protein, 15 of which occur in the receptor binding domain (RBD), plus three deletions and one triple-residue insertion in NTD. Combined vaccines elicit ultrapotent neutralizing antibodies and gain protected humoral immunity against multiple variants, including Beta and Omicron, as determined by single memory B cell sequencing [20]. Invasion by Omicron or other variants recalls the humoral responses of the single memory B cell-derived antibody repertoire, conferring secondary protection. However, neither mechanisms nor signal carriers have been fully elucidated.

## 5. Perspective: Beyond PPSV 23-Valent Unconjugated Pneumococcal Vaccine to Venture to Cancer Vaccine

Subclonal evolution spatiotemporal tracking system guides diagnostic and treatment. 

It feels surreal going from caring for the initial onslaught of patients last spring to helping alleviate human suffering through vaccinations. Some readers pointed out that administering a plain Polyvalent polysaccharide vaccine is highly relevant to a messenger RNA vaccine, mRNA-1273, to prevent infection with SARS-CoV-2 after the first vaccination [4] and the second vaccination [5]. The mRNA-1273 vaccine can neither prevent the infection nor spread of SARS-CoV-2, but it can prevent the disease after infection or reduce the severity and damage of the disease. Generally speaking, this feature is why many people need to be vaccinated to achieve herd immunity through vaccines. The more infectious a virus is, the higher the vaccination rate is required. Another line of pioneering work has been “cancer drug resistance in the context of caveola-associated molecules and tumor stroma,” which sheds new light on developing new treatments [110].

These findings suggest that classic vaccines do not block viral entrance but neutralize viral load-derived toxicity. The breathtaking knowledge of COVID-19 vaccines sheds new light on such a grand old field. We suggest a new mechanism by which the dual targets of both Cav-1 and SARS-CoV-2 spike proteins might block the viral entrance and neutralize viral load, minimizing the immune reaction against viral attacks and clearing out the host’s viral survival. Thus, bacteria can activate the host’s immunity to antitumors. For example, Kalaora et al. [111] report that certain bacteria can invade melanoma cells, as evidenced by the presence of bacterium-derived peptides. T cells recognize and are activated against these peptides [112]. T lymphocytes can identify these bacterial peptides to boost immunity. All of these relate to tumor immunity.

Therapeutic intervention of the tumor microenvironment (TME) with natural products and re-purposed pharmaceuticals illustrates the mechanisms by which exosomes trigger inflammation, leading to immune evasion and tumor progression, and remodeling of the tumor microenvironment, remarkably through EV-mediated signal pathways [113].Tumor-derived exosomal non-coding RNAs (ncRNAs), miRNAs, lncRNAs, circRNAs, and other genes, such as PTEN, PI3K/Akt, and STAT3, regulate molecular signaling pathways in cancer progression. Thus, the therapeutic efficacy, diagnosis, and prognosis of EVs in cancer patients remain to be elucidated.

Conclusion: we indicate that polysaccharides induce memory B cells that are indeed different from the mode of action of protein antigens. Polysaccharide-specific immunoglobulin G (IgG) governs memory B cells that act as T-independent type II immune responses in naive B cells sensitive only to polysaccharides. We offer an opinion regarding caveolae-mediated extracellular vesicle (CMEV) signaling in the administration of the plain Polyvalent polysaccharide vaccine against *Streptococcus pneumoniae* (pneumococcus), which appears to be highly relevant to the primary vaccine against SARS-CoV-2 and its booster dose. 

## Figures and Tables

**Figure 1 pharmaceutics-14-02653-f001:**
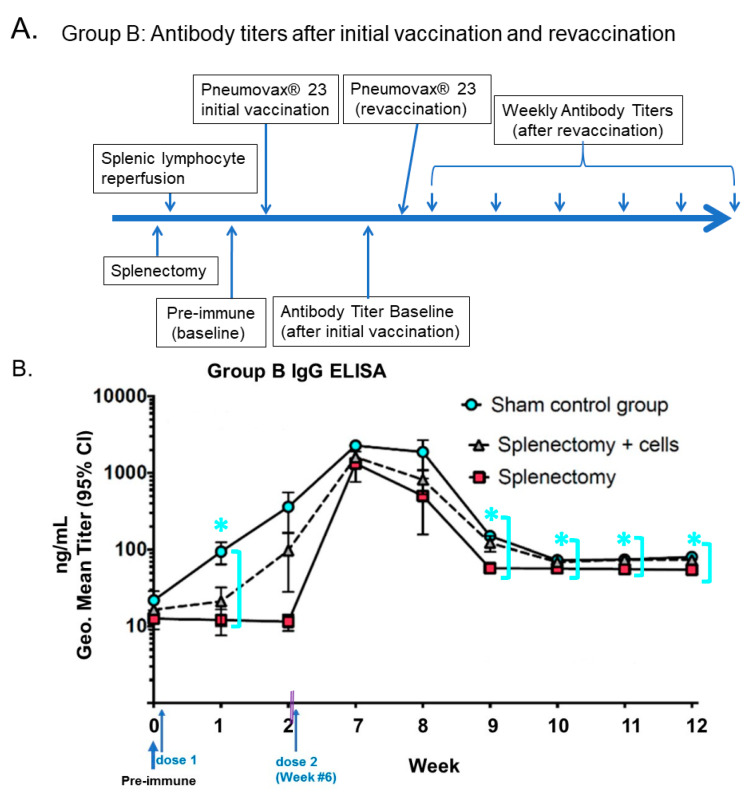
Humoral antibody-mediated immune response to PPSV23 vaccination in a mouse model following splenectomy. (**A**) Timelines of PPSV23 vaccination. (**B**) Antibody IgG titers of Balb/C mice with the repeat vaccination (6 weeks) and tested against polysaccharides conjugated to ELISA plates. (* signals the statistically significant, *p* < 0.001, sham control subgroup being compared with −SL subgroup for weeks 10, 11, 12), as sky-blue circles compared with red squares (splenectomy), as shown with mean ± S.D. concentrations (ng/mL titers) and 95% confidence intervals as measured by ELISA with Log10 scale). Doubled-purple-lines defined a time gap from 2 weeks to the repeat vaccination at 6 weeks. (“Geo.”: it stands for geometric mean.). (Refer to [14] for details). [Figure reproduced License 4.0 (CC BY-912)].

**Figure 2 pharmaceutics-14-02653-f002:**
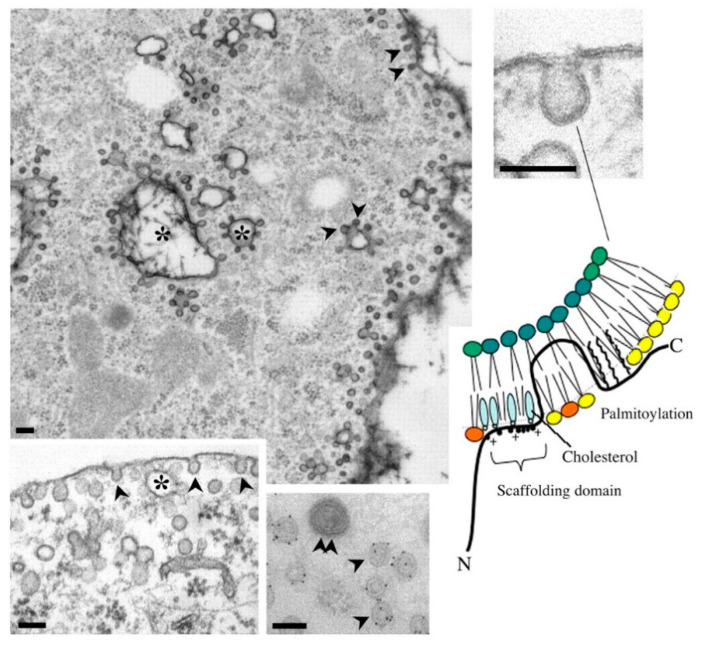
Cav-1-mediated signal transduction through the Caveolin-Scaffolding-Domain (CSD) [39] that interacts with a CSD-interacting motif (CIM) among the partner receptors [40] (adapted from [41]. Arrowheads indicate caveolae organelles. The “+” indicates the conserved amino acid residues of caveolin-interacting motifs (CIM): ΦXΦXXXXΦ and ΦXXXXΦXXΦ, where Φ is an aromatic residue (Trp, Phe, or Tyr) [40]. [Figure reproduced License 4.0 (CC BY-912)].

**Figure 3 pharmaceutics-14-02653-f003:**
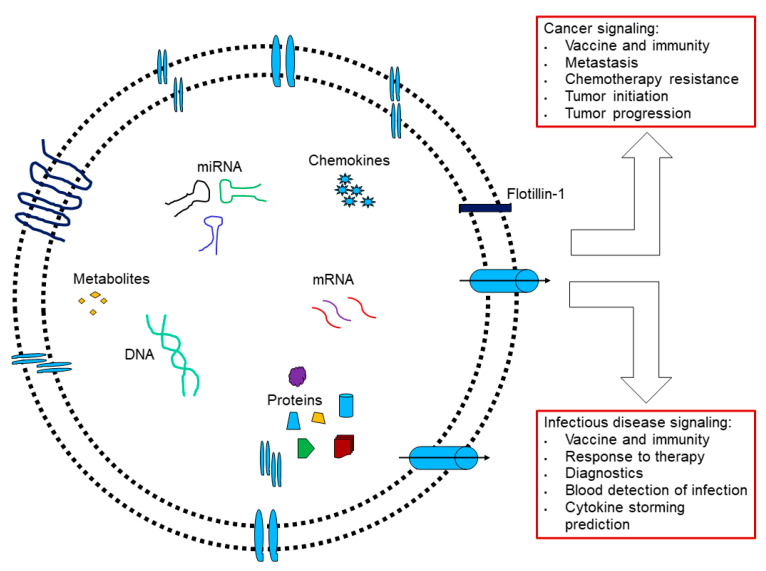
Extracellular vesicles are involved in cancer and infectious diseases. (Adapted from [42]).

**Figure 4 pharmaceutics-14-02653-f004:**
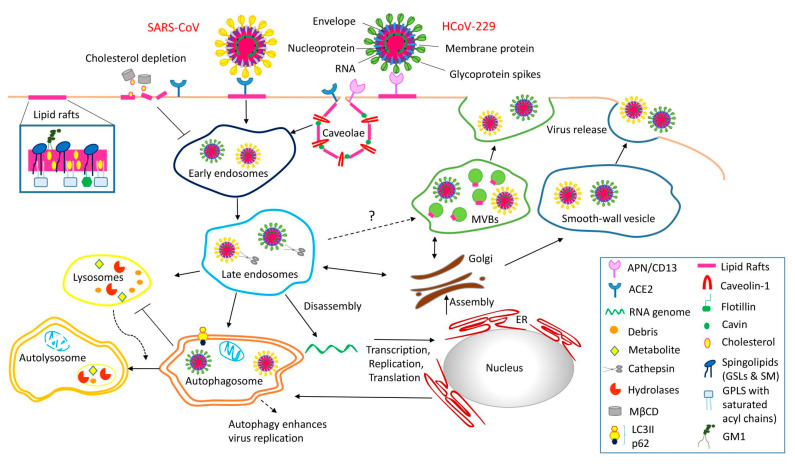
SARS-CoV-2 infection is tightly regulated spatiotemporally. A viral spike glycoprotein bound to the host receptor (ACE2, APN/CD13) of lipid rafts/caveolae for viral entry triggers the downstream mechanism of pathogenesis and immunity (adapted from [41]. Alternatively, viruses hijack autophagy machinery for viral replication. Third, we speculate that coronavirus might utilize multivesicular bodies (MVBs) and take advantage of the exosomal pathway for egress (adapted from [41]. Caveolae-budded EVs are potential hijackable gates in cell communication of caveolin-related diseases named caveolinopathies. Note that the question mark indicates an alternative pathway as speculated for coronavirus to utilize multivesicular bodies (MVBs) and to take advantage of the exosomal pathway for egress. [Figure reproduced under Creative Commons Attribution-NonCommercial License 4.0 (CC BY-912)].

**Figure 5 pharmaceutics-14-02653-f005:**
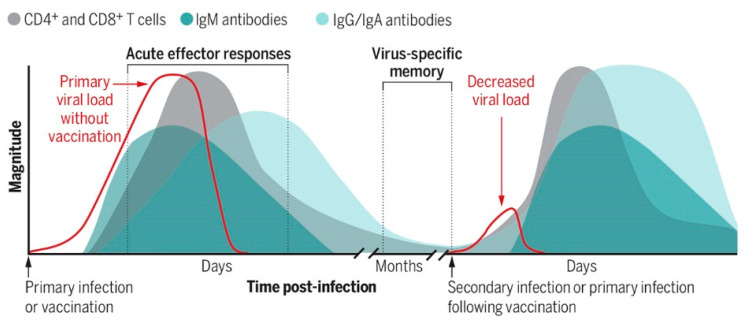
Viral invasion triggers the adaptive immune system.

**Figure 6 pharmaceutics-14-02653-f006:**
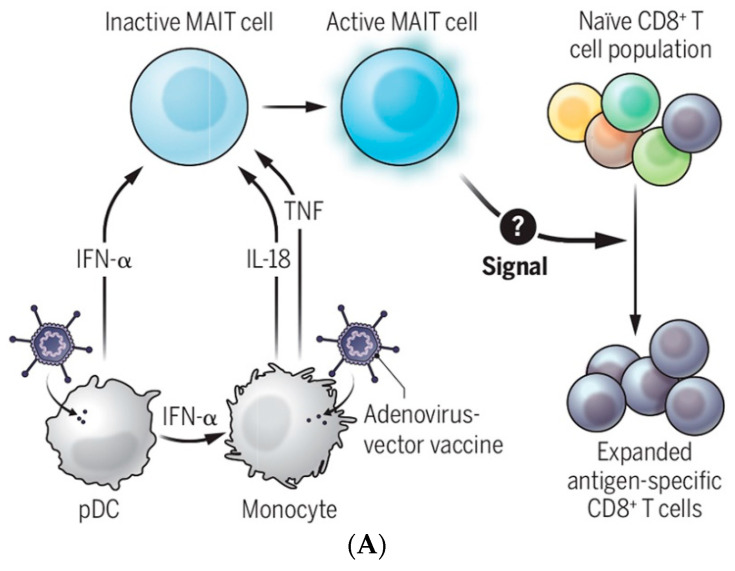
Communicating vaccine immunogenicity derived from three EV production modes associated with parasitic infection. (**A**) Adenoviral vaccine acts on pDCs and monocytes to produce IFN-α, IL-18, and TNF to activate MAIT cells, which might produce EVs (our hypothesis) to expand vaccine antigen-specific CD8+ T cells [73]. We speculate that the signal in question (with a question mark) is carried by exosomes or microvesicles, including caveolae-mediated extracellular vesicles (EVs), as proposed here. (**B**) Three EV production modes are associated with parasitic infection [23]. [Figure reproduced from [47] with the permission of Science—the AAAS Publishing].

## Data Availability

Not applicable.

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
