# Peer review of "Caveolae-Mediated Extracellular Vesicle (CMEV) Signaling of Polyvalent Polysaccharide Vaccination: A Host–Pathogen Interface Hypothesis†"

_pharmaceutics, 2022, doi:10.3390/pharmaceutics14122653_

Round 1

Reviewer 1 Report

There are some interesting opinions about caveolae-mediated EV-immunity from polyvalent polysaccharide in this article. In fact I'd rather classify this article as a hypothesis but not a review. The thoughts in this paper should be verified by more published literature to become a review.

Author Response

attached PDF

Reviewer 2 Report

The overall manuscript looks good, thus recommend for publication.

Author Response

attached PDF

Reviewer 3 Report

In this manuscript, the authors offer an opinion on the administration of the plain Polyvalent polysaccharide vaccine against PPSV23 which appears to be highly relevant to the primary vaccine against SARS-CoV-2 and as a booster dose.

 The authors indicated that there was a debate concerning whether the Polyvalent polysaccharide vaccine (PPSV23) by splenocyte reinfusion, induced  primary and secondary immune responses or allowed humoral antibody responses to initial vaccination and revaccination. Thus, the authors do not support a secondary immune reaction, but an antibody response measured through IgG titers after revaccination. They also indicate that they are not sure if these lower but present IgG levels against the pneumococcal antigen is protective clinically  or are equally low in all groups because of the phenomenon of "hyporesponsiveness" seen after repeat polysaccharide vaccine challenge. Also, they indicate that polysaccharides create memory B cells that are indeed different from mode of action of protein antigens. A polysaccharides-specific immunoglobulin G (IgG) governs the memory B cells that act as T-independent type II immune responses in naive B cells sensitive only to polysaccharides.  

In addition, the authors decently reviewed  the literature and offered a new mechanism based on the caveolae memory extracellular vesicles (CMEVs), by which polysaccharides mediate prolong and sustain vaccination and immunotherapy.  

The authors conclude and explain that the dual targets onto both Cav-1 and SARS-CoV-2 spike protein may block the viral entrance and neutralize the viral load, which minimize the immune reaction against viral attacks and clear out the viral survival of the host. They finally propose a new uniform hypothesis for all vaccination strategies, i.e., caveolae-mediated extracellular vesicles mediate vaccine memory.

 This manuscript is generally well  written, though, some of the writing, although grammatically correct and perfect (but very complex), may obscure some of the scientific meanings. The authors while presenting their excellent immunological scientific opinion, have also answered queries and responses made by readers to their original statements published in their previous work  on the: “ Autologous splenocyte reinfusion improves antibody-mediated immune response to the 23-valent Pneumococcal polysaccharide-based vaccine in splenectomized mice”, published in Biomolecules. 2020, 10(704), 701-716.

The following are few examples that shows the complexity of the English language, these are few examples, and the authors should try to simplify the complexity of their sentences.

1.In page 2 of your pdf, lines 84-85,  please correct as shown:

This report showed that polysaccharides create memory B cells that differ from the protein antigens.

2. In page 2,  lines  87-91, you have written the following:

Polysaccharide antigens fail to be recognized and bound by the T cell antigen receptor complex; however, neither do we know  polysaccharides are TI-2 antigens stimulate T cells, nor by T cell help, but by non-T cells (e.g., the NK cell) to interact with the B cells either directly or indirectly via cytokine production (as IL-3, GMCSF, and IFN-gamma) (Mond et al., 1995).

Query: Perhaps written in proper English language, however this sentence is cryptic! Please rewrite more clearly

3. In page 6, lines 234-236, you have written the following:

A new theory was that cavins garnish a lipid domain for caveolin into the bilayer to promote caveola formation, which is a dynamic "metastable membrane domain that can be readily disassembled both in response to cellular stress and during endocytic trafficking.

Query: The use of this obtuse poetic or Shakespearean  language (cavins garnish!) only serves to distort the meaning of this sentence. When you are writing an opinion, you need to keep the sentence structures simple so that people who are not experts in this field could understand what the authors means!

Author Response

attached PDF
